# Innate Immunity in Cardiovascular Diseases—Identification of Novel Molecular Players and Targets

**DOI:** 10.3390/jcm12010335

**Published:** 2023-01-01

**Authors:** Wolfgang Poller, Bettina Heidecker, Enrico Ammirati, Andreas W. Kuss, Ana Tzvetkova, Wolfram C. Poller, Carsten Skurk, Arash Haghikia

**Affiliations:** 1Department of Cardiology, Campus Benjamin Franklin, Charité—Universitätsmedizin Berlin, Corporate Member of Freie Universität Berlin, Humboldt-Universität zu Berlin, Berlin Institute of Health, 12200 Berlin, Germany; 2Berlin-Brandenburg Center for Regenerative Therapies (BCRT), Charité—Universitätsmedizin Berlin, 12200 Berlin, Germany; 3German Center for Cardiovascular Research (DZHK), 12200 Berlin, Germany; 4De Gasperis Cardio Center and Transplant Center, Niguarda Hospital, 20162 Milano, Italy; 5Department of Functional Genomics, Interfaculty Institute of Genetics and Functional Genomics, 17475 Greifswald, Germany; 6Institute of Bioinformatics, University Medicine Greifswald, 17487 Greifswald, Germany; 7Cardiovascular Research Institute, Icahn School of Medicine at Mount Sinai, New York, NY 10029, USA; 8Berlin Institute of Health, Charité-Universitätsmedizin Berlin, 10117 Berlin, Germany

**Keywords:** cardiovascular diseases, immunology, innate immunity, immunogenetics, noncoding genome, RNA interference, antisense therapeutics, gut microbiome, neuroimmunology

## Abstract

During the past few years, unexpected developments have driven studies in the field of clinical immunology. One driver of immense impact was the outbreak of a pandemic caused by the novel virus SARS-CoV-2. Excellent recent reviews address diverse aspects of immunological re-search into cardiovascular diseases. Here, we specifically focus on selected studies taking advantage of advanced state-of-the-art molecular genetic methods ranging from genome-wide epi/transcriptome mapping and variant scanning to optogenetics and chemogenetics. First, we discuss the emerging clinical relevance of advanced diagnostics for cardiovascular diseases, including those associated with COVID-19—with a focus on the role of inflammation in cardiomyopathies and arrhythmias. Second, we consider newly identified immunological interactions at organ and system levels which affect cardiovascular pathogenesis. Thus, studies into immune influences arising from the intestinal system are moving towards therapeutic exploitation. Further, powerful new research tools have enabled novel insight into brain–immune system interactions at unprecedented resolution. This latter line of investigation emphasizes the strength of influence of emotional stress—acting through defined brain regions—upon viral and cardiovascular disorders. Several challenges need to be overcome before the full impact of these far-reaching new findings will hit the clinical arena.

## 1. Introduction

Multiple excellent reviews have addressed diverse important aspects of immunological research into cardiovascular diseases during the past few years. In this review, we specifically focus on preclinical and clinical studies which have provided unexpected insights by taking advantage of recent state-of-the-art molecular genetic and virological technologies, ranging from clinical genome-wide transcriptome mapping and variant scanning to optogenetics and chemogenetics. Due to intense worldwide efforts in these fields during the past years, the present review cannot be comprehensive, but instead tries to convey an up-to-date perspective on promising developments which may shape research at the crossroads of cardiology–immunology-neurology (Figure 1). Whereas advanced technologies are often confined to applications in basic research, we focus here on those with already proven or upcoming use in the clinical arena.

## 2. Molecular Immunogenetics of Cardiovascular Diseases

While molecular genetic methods have been extensively employed in cardiovascular research for decades, only the rather recent advent of comprehensive and still affordable mutation/variant scanning tools ready for day-to-day clinical practice have significantly enhanced their clinical impact. As with all other topics discussed in this review, availability of practice-ready analytical tools is not simply a gradual step forward, but the critical threshold before widespread relevance for clinical medicine may be achieved. This degree of technological evolution may well take decades, as highlighted in recent reviews [1,2] covering therapeutics based on noncoding RNAs (ncRNAs) and related nucleic acids. Once achieved, however, this defines the quantum leap from a “promising” to a “revolutionary” medical development.

### 2.1. Novel Insights into the Role of Inflammation in Human Cardiomyopathies

Despite multiple obstacles to unequivocal detection of myocardial inflammation, it has been long known that myocardial inflammation [3,4,5,6,7,8,9,10,11,12] occurs in association with multiple types of ‘genetic‘ [13,14,15,16,17] and ‘non-genetic‘ [18,19] cardiomyopathies. Figure 2 provides an overview of currently known cellular receptors and signaling pathways linking innate immunity with cardiovascular diseases (CVD). Pattern recognition receptors (PRRs) may be associated with the cell membrane (Toll-like receptors, C-type lectin receptors), located within the cytosol (NOD-like receptors, RIG-like receptors), or in specialized intracellular compartments (endosomes) particularly relevant during viral infections (TLRs 3, 7, 8, and 9). Our mechanistic understanding of the highly complex multiscale signaling network of innate immunity is still incomplete. Figure 2 displays well-established classical players of the innate immune system to which, however, multiple novel components, such as immunomodulating miRs or lncRNAs, need to be added (Figure 3). For an in-depth analysis of current knowledge on classical players of cardiovascular immunity, we refer the reader to Jaen et al. 2020 [20], and for an overview on CVD inflammation in general, to a comprehensive review series [21].

Recent clinical studies employing highly advanced but clinical practice-ready genetic diagnostics [4,24,25,26] have revealed a significant direct impact of myocardial inflammation upon the induction of life-threatening arrhythmias. These studies investigated patients with well-defined genetic anomalies identified by large-scale mutation scanning. This research has so far led to the identification of a novel innate immune sensor (SCN5A sodium channel), and of hitherto unknown innate immune triggers (mutant desmosomal proteins) with significant implications for the clinical management of affected patients. 

Thus, a missense variant E1295K of the sodium channel gene SCN5A was found to be associated with recurrent ventricular fibrillation and myocardial inflammation [25]. In that study, an immunosuppressive therapy course with prednisolone led to stabilization of cardiac rhythm and marked clinical improvement. SCN5A encodes sodium channel α-subunit responsible for action potential initiation and conduction of electrical stimuli through the heart. SCN5A was initially assumed to be exclusively expressed in the myocardium, but recently a SCN5A splice variant was found to activate antiviral innate immune signaling [27]. Pioneering work revealed that SCN5A modulates myelin degradation by macrophages in multiple sclerosis (MS) and that overexpression of the macrophage SCN5A variant in mice protects against murine experimental MS [28]. Endosome-associated SCN5A variants thus emerged as novel innate immune sensors, indicating that patients so far classified as ‘pure‘ myocardial ion-channel disease cases may carry independent ‘immunologic‘ risk through hitherto-neglected anomalous function of their mutant ion channels. Another study taking advantage of high throughput mutation scanning found familial recurrent myocarditis to be triggered by exercise in patients with a truncating variant of the desmoplakin gene [24]. This work illustrates the potential of advanced genetics in combination with state-of-the-art clinical myocardial diagnostics not only to improve clinical practice, but also to reveal unexpected pathogenetic processes. A third paper [4] investigating acute myocarditis associated with desmosomal gene variants (DGVs) found a strong adverse impact of DGV-associated inflammation upon ventricular arrhythmogenesis and survival. 

Unfortunately, there are multiple obstacles to unequivocal clinical recognition of myocardial inflammation as a precipitating factor for life-threatening arrhythmias or the development of terminal heart failure. Even in narrowly focused patient cohorts for whom genetic predisposition is suspected, e.g., by family history or direct detection of pathogenic variants [4,24,25], recognition of myocardial inflammation requires in-depth diagnostic work-up (cardiac magnetic resonance imaging—cMRI; endomyocardial biopsy—EMB; positron emission tomography-computed tomography—PET/CT), which may be unfeasible in many cases. Many of these patients may carry implantable cardioverter-defibrillators (ICDs), often preventing reliable cardiac MRI diagnostics, and diagnostic accuracy of EMB may be limited by sampling error. Even *post mortem* examination would be unable to detect transient bouts of inflammation during the sometimes decades-long course of cardiomyopathies, unless inflammation persists until the time of death. 

It is most likely that only a minute fraction of all inflammation-triggered arrhythmic events or critical heart failure progression is currently detected in routine clinical practice, generating a blind spot regarding the potential value of anti-inflammatory treatments in these contexts. The above studies employing state-of-the-art genetic tools strongly suggest to further evaluate the hypothesis of independent immunologic risk in major cohorts of patients carrying SCN5A or desmosomal gene variants. Conduction of such studies is challenging indeed and requires multicentric cooperation between experienced centers, but given the often young age of affected patients and possible life-saving impact, the effort appears warranted. For an in-depth discussion regarding design and key problems of immunomodulating treatment trials in human cardiomyopathies, we may refer the reader to a comprehensive recent review [29].

### 2.2. Common Gene Variants Affecting Antiviral Response and Myocardial Disease

To illustrate that not only rare genetics variants, but also rather common genetic polymorphisms may significantly influence the immune response and associated cardiovascular diseases, we briefly discuss here a number of studies on the forkhead transcription factor Foxo3 [30,31]. Foxo3 is involved in cell cycle regulation, apoptosis, oxidative stress, angiogenesis, and immunity. The immune-modulatory function of Foxo3 in adaptive immune responses has been elucidated to some extent. Foxo3 contributes to maintenance of T cell tolerance and quiescence, and the differentiation of regulatory T cells is regulated by the transcription factor [32,33,34]. Moreover, Foxo3 maintains neutrophil vitality in models of neutrophil inflammation [35]; plays an important role in cardiac hypertrophy [36], cardiomyocyte survival [37], cell differentiation, and remodeling [38]; and provokes resistance to oxidative stress in cardiac fibroblasts [39].

Remarkably, single nucleotide polymorphisms (SNPs) of the *FOXO3* gene are associated with longevity [40,41] and low prevalence of cardiovascular diseases in diverse populations. These impressive initial studies have triggered a broad spectrum of research into translational aspects of Foxo3. Regarding the immune response, a human SNP in *FOXO3* is associated with increased risk for malaria, but a milder course in patients with autoimmune disease [42]. Within the cardiovascular field, FOXO3 has gained interest with regard to virus-triggered myocarditis [43] which is associated with high mortality and is an important cause for the need for heart transplantation. It is not well understood how the immune system recognizes and controls myocardial coxsackievirus B3 (CVB3) infections [44,45], but murine studies suggest NK cells play a critical role in viral clearance and host survival. Consistent with this, a translational experimental and clinical study by Loebel et al. [43] found an association of the FOXO3 SNP rs12212067 with human NK cell function, and also the clinical outcome in patients with virus-positive inflammatory cardiomyopathy. These findings thus corroborate prior evidence from animal studies. Importantly, this study suggests a dual role of FOXO3 genetic variants. While enhanced FOXO3 activity associated with rs12212067 may be protective in chronic inflammation, e.g., cancer and cardiovascular disease, it appears to be disadvantageous to control acute viral infection.

### 2.3. Novel Immune Players from the Human Noncoding Genome 

Similar to the above-considered evolution of molecular genetic diagnostics, decades passed from the discovery that about 99% of the human genome does not encode proteins, but instead generates a broad spectrum of noncoding RNAs (ncRNAs), many of whom are involved in the immune response [46,47,48,49,50,51,52,53,54,55,56,57,58,59,60,61,62], until finally successful clinical exploitation of ncRNAs and of novel drugs developed using them as blueprints was achieved [1,2]. Across the entire spectrum of medical disciplines, it has been ascertained that the noncoding genome plays a key role in genetic programming and gene regulation during development, as well as in health and disease.

Figure 3 displays the general structure and some key features of the human noncoding genome and epigenome, as well as suggested novel therapeutic targets therein. About 99% of the human genome does not encode proteins, but instead give rise to a broad spectrum of ncRNAs with regulatory and structural functions. While most questions regarding the overall clinical impact of the noncoding genome are still unanswered, experimental and first clinical data suggests a number of microRNAs (miR-17-92 cluster, miR-21, miR-142, miRNA-146a, miR-155, miR-181, miR-223) and long noncoding RNAs (lncRNAs) may constitute therapeutic targets in cardiovascular diseases. Experimental studies also shed first light upon therapeutic targeting of the epigenome in heart failure, but it should be emphasized [2] that epigenetic modifications likely result in pleiotropic actions, making clinical translation particularly challenging.

In order to provide a fully balanced state-of-the-art assessment of the technically highly demanding and rapidly evolving field of human genome research, we briefly address some current fundamental knowledge gaps regarding the molecular workings of diverse ncRNA species within intact fully functional cells. While the biosynthetic pathways and interactions of many miRs and RNAi-inducing siRNAs have been extensively characterized, this is not the case for the highly diverse species summarized under the rather unspecific term ‘long noncoding’. This huge pool encompassing many thousands of transcripts is in fact essentially unexplored with regard to mechanisms, as well as biological relevance in health and disease. Different lncRNAs may modulate nuclear ultrastructure (e.g., *MALAT1*, *NEAT1*), create RNA–protein interaction surfaces, or target RNA–protein complexes to specific genomic regions, making obvious that a more straightforward classification is needed in the future. While reductionist analysis in cell-free or other simple systems may reveal molecular insights, this does not yet reveal biological function in an intact differentiated cell or in vivo. Recent critical reviews [63,64] have pointed out that essentially each lncRNA needs to be analyzed per se because of their diversity, and they therefore suggest to focus research upon species fulfilling diverse criteria of biological relevance in humans.

Thus, Ponting and Haerty [64] propose research prioritization of human lncRNAs which (1) display sequence conservation and are transcribed in other mammals; (2) are abundant in at least one type of primary cells (although some bona fide lncRNA are expressed at low levels); (3) show specific subcellular localization suggesting functional hypotheses; (4) interact with defined other molecules (although lncRNA with ascertained function, e.g., *NEAT1,* are functionally dependent on proper formation of multi-component (RNAs, proteins) complexes. With caution, one may also incorporate evolutionary perspectives into these considerations. It would appear that evolutionary spread across divergent animal species suggests a greater likelihood that it plays a role in human biology. Nevertheless, lncRNAs of very recent evolutionary origin (primates) may well convey significant functional advantage and pathogenetic relevance.

**Figure 3 jcm-12-00335-f003:**
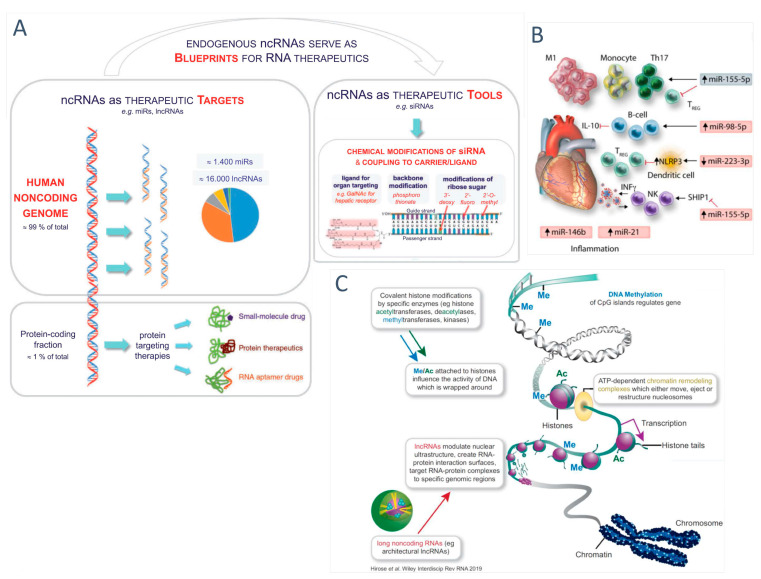
Key features of the human noncoding genome and epigenome, and novel therapeutic targets therein (modified from Poller et al. Eur. Heart. J. 2018 [1] and 2020 [2] by permission). (**A**) Noncoding RNAs are therapeutic targets and blueprints for therapeutic tools. (**B**) Inflammation-associated miR targets. (**C**) Epigenome structures and modifiers. *Panel A*: About 99% of the human genome does not encode proteins, but are transcriptionally highly active and give rise to a broad spectrum of ncRNAs with regulatory and structural functions. The observation of a steeply increasing fraction of noncoding RNAs (ncRNAs), in contrast to the modest increase in the number of protein-coding genes during evolution from simple organisms to humans, suggests a major role of ncRNAs in humans. ncRNAs, including microRNAs (miRs), small interference RNAs (siRNAs), or long noncoding RNAs (lncRNAs), are involved in the maintenance of cellular homeostasis and the innate immune response to tissue injury or infection. *Panel B*: While many questions regarding the overall impact of the noncoding genome upon human health and disease are still unanswered [63,64,65], available experimental and clinical evidence so far suggests a limited number of miRs (miR-17-92 cluster, miR-21, miR-142, miRNA-146a, miR-155, miR-181, and miR-223) [66] and lncRNAs [1,2,67] may constitute therapeutic targets in cardiovascular diseases. For each individual target, however, this depends on ensuring efficient drug targeting and clinical safety [68]. *Panel C*: Experimental studies [69] shed first light upon therapeutic targeting of the epigenome epigenetics in the context of heart failure. While epigenetic modification is a highly interesting field of research, it should be emphasized [2] that epigenetic modifications are likely to result in pleiotropic actions, making clinical translation particularly challenging *(modified from [1,2] by permission of Eur. Heart J.).*

### 2.4. Novel Nucleic Acid Therapeutics Targeting Conventional Protein-Coding Genes and Noncoding RNA Targets

Within the cardiovascular field, multiple early experimental studies [70,71,72,73,74,75,76,77,78,79] showed that certain ncRNAs (miRNAs) are regulators of cardiovascular pathogenesis in animal models. This of course immediately suggested they might have potential to improve diagnostics and could possibly even be developed into novel therapeutics. The road to in-depth understanding of the molecular workings of at least a few of the numerous ncRNA classes, and the development of sophisticated bioengineered nucleic acid drugs [70,80,81,82,83,84,85,86,87,88,89,90] safe and efficacious for clinical applications, took two decades, counting from early experimental work to the first clinically successful trials. 

Figure 4 provides an overview on novel nucleic acid therapeutics targeting conventional protein-coding genes or noncoding RNA targets with relevance in cardiovascular disorders. Particularly advanced is the development of RNA interference (RNAi) drugs, which use recently discovered pathways of endogenous short interfering RNAs and are becoming versatile tools for efficient silencing of protein expression. Cardiovascular RNA drugs may address multiple organ systems. Of note, they need not address heart or vasculature directly, but instead, primarily liver-targeted RNA drugs are the currently most successful development in the cardiovascular field.

Thus, pioneering clinical studies include RNAi drugs targeting liver synthesis of PCSK9, resulting in highly significant lowering of LDL cholesterol or targeting liver transthyretin (TTR) synthesis for treatment of cardiac TTR amyloidosis. Further novel drugs mimicking actions of endogenous ncRNAs may arise from exploitation of molecular interactions not accessible to conventional pharmacology. For a more in-depth coverage of the enormously challenging bioengineering, safety, and regulatory hurdles to be overcome towards clinical therapy during the past decades, we may refer the reader to comprehensive recent reviews [1,2,91].

Whereas a series of ground-breaking clinical trials [68,84,88,89,92,93,94,95,96,97,98,99,100,101,102,103,104,105,106,107,108,109,110,111,112] has provided definite evidence of the therapeutic potential of RNA interference and antisense drugs for cardiovascular disorders, the inclusion of ncRNA profiling into the clinical diagnostic process and prognosis assessment is less conclusive so far. Still, it may significantly contribute to optimizing patient care in selected complex or otherwise equivocal cases. Thus, rapid diagnosis of life-threatening idiopathic giant cell myocarditis and cardiac sarcoidosis is significantly improved by myocardial gene expression profiling [113]. In another disease with highly variable clinical course and outcome, human enterovirus cardiomyopathy, differential cardiac microRNA profiling helps to predict the clinical course and the need for antiviral therapy [114]. As another example, circulating exosomal microRNAs predict functional recovery after interventional repair of severe mitral regurgitation [115]. Of course, it needs to be emphasized that the definitive establishment of predictive or differential diagnostic expression profiles requires confirmation by several independent clinical studies. This has been rarely achieved so far [1,2], but the available evidence does suggest significant clinical potential for selected clinical settings [1,2,116,117,118,119].

### 2.5. Continuously Emerging New Levels of Complexity of the Human Genome

The more recent history of human genetics is characterized by several revolutionary discoveries [1,2]. After the very important sequencing of the entire genome of humans (and meanwhile of a vast number of other species), it became apparent that 99% of the ascertained sequence does not encode proteins. This already suggested that this huge noncoding fraction plays truly critical and so far only very incompletely understood roles for the proper functioning and environmental adaptations of individuals, but for evolution itself through advanced species. Furthermore, prenatal influences, as well as environmental factors, are known to alter the genome through epigenetic mechanisms or imprinting. An individual’s genome is on the one hand rather fixed at DNA level (except somatic mutations/recombinations), but its functional status may still be significantly and durably altered by the environment via epigenetics. One may safely assume these complexities of evolutionary advanced genomes convey a distinct survival benefit; otherwise, they would not have become commonplace in higher species. In the following, we consider further data suggesting that certain genomic regions give rise to multiple immunomodulatory transcripts interacting with each other in a way reminiscent of integrated electronic circuits to generate an optimized responses to complex inputs. All hitherto known functional levels of the genome—from DNA editing [88,105,106] to epigenome-targeting “epi-drugs“ [107,120,121,122,123,124,125,126]—have already been investigated with regard to possible therapeutic potential. 

To illustrate the multi-level functional integration of major strands of the human genome, we invoke the evolutionarily conserved *NEAT1-MALAT1* cluster encountering a high level of interest in both cardiovascular medicine and oncology. In the cardiovascular field, suppression of lncRNA NEAT1 was observed in circulating immune cells of post-myocardial infarction (MI) patients. Mice lacking the lncRNAs *NEAT1 or MALAT1* displayed immune disturbances affecting monocyte-macrophage, as well as T cell differentiation, rendering the immune system highly vulnerable to stress stimuli, thereby promoting the development of atherosclerosis. Uncontrolled inflammation is also a key driver of multiple other diseases. 

The human *NEAT1-MALAT1* gene cluster generates large noncoding transcripts remaining nuclear, while tRNA-like transcripts (mascRNA, menRNA), enzymatically generated from these precursors, translocate to the cytosol (Figure 5). *NEAT1^-/-^ and MALAT1^-/-^* mice display massive atherosclerosis and vascular inflammation [127,128,129,130]. A recent study found that these tRNA-like molecules are critical components of innate immunity. They appear as prototypes of a new class of noncoding RNAs distinct from others (miRNAs, siRNAs) by biosynthetic pathway (enzymatic excision from lncRNA precursors) and intracellular kinetics. 

CRISPR-generated human ΔmascRNA and ΔmenRNA monocytes/macrophages display defective innate immune sensing, loss of cytokine control, imbalance of growth/angiogenic factor expression impacting upon angiogenesis, and altered cell–cell interaction systems. Antiviral response, foam cell formation/oxLDL uptake, and M1/M2 polarization, are defective in ΔmascRNA and ΔmenRNA macrophages, defining the tRNA-like molecules’ first described biological functions [131]. This CRISPR-Cas9-based cell biological study revealed that menRNA and mascRNA represent novel components of innate immunity arising from the noncoding genome.

For future translational studies, the approach to investigate menRNA and mascRNA directly suggests new avenues, since their highly dynamic levels may be more closely related to clinical parameters and clinical course than those of their nuclear precursors. The observed foam cell formation/oxLDL uptake defects of ΔmenRNA macrophages suggest menRNA and mascRNA to be involved in atheropathogenesis. They may have value as therapeutic targets for pharmacological intervention, as they are more easily accessible than their complex nuclear-located precursor molecules. A prior observation that recombinant mascRNA [130] abolishes virus replication in cardiomyocytes in fact suggests therapeutic potential of mascRNA and menRNA-targeting interventions.

From a more general point of view, the *NEAT1-MALAT1* region appears as archetype of a functionally highly integrated RNA processing system [131]. Figure 5 summarizes knowledge from a number of ground-breaking studies from several research groups and illustrates the surprising complexity and many unknowns of this genomic region. We propose to view the *NEAT1-MALAT1* region as a biological type of highly integrated circuit deeply involved in the control of innate immune sensing and cell–cell interactions.

Two other recent studies illustrate the power of state-of-the-art genetics to reveal fundamentally new insights into immune system evolution [132] and large-scale human evolution in general [133]. Thus, a protective HLA extended haplotype was found to outweigh the major COVID-19 risk factor inherited from Neanderthals in the Sardinian population [132]. Another large study [134] conducted multilocus genotyping of SARS-CoV-2 genomes sampled globally and found evidence of the majority of SARS-CoV-2 infections in 2020 and 2021 caused by genetically distinct variants that likely adapted to local populations. For further information on novel insights into genome structure and function, we may refer the reader to a series of recent landmark papers and websites [64,135,136,137,138,139,140,141,142,143,144,145,146,147,148,149,150].

Beyond the developments as reviewed in Section 2.3, a further level of complexity, i.e., the human epigenome and its lifetime dynamics in health and cardiovascular disease [91,151,152,153,154,155,156,157,158,159,160], emerged as holding promise for possible therapeutic exploitation [120,121,122,123,124,125,126,161,162]. At the present time, however, the impact of epigenetic drugs upon clinical medicine is still limited as compared to the nucleic acid drugs discussed above. 

## 3. Current Expansion of Virological Research in the Cardiovascular Field

### 3.1. Impact of Viral Infections upon the Cardiovascular System

Infection of the myocardium with cardiotropic viruses is one of the main causes of myocarditis and acute and chronic inflammatory cardiomyopathy (DCMi). However, viral myocarditis and subsequent dilated cardiomyopathy is still a challenging disease to diagnose and to treat and is therefore a significant public health issue globally [163]. Advances in clinical phenotyping and thorough molecular genetic analysis of intramyocardial viruses and their activation status have incrementally improved our understanding of molecular pathogenesis and pathophysiology of viral infections of the heart muscle. To date, several cardiotropic viruses have been implicated as causes of myocarditis and DCMi. These include, among others, classical cardiotropic enteroviruses (Coxsackieviruses B) [44,164], the most commonly detected parvovirus B19 [165], human herpes virus 6 [166,167], and hepatitis C virus (HCV) [168,169]. An entirely unwelcome newcomer is the respiratory virus which has triggered the worst pandemic since a century ago, SARS-CoV-2, whose involvement and impact in viral cardiovascular disease is under scrutiny [170,171,172,173,174,175]. Despite extensive research into the pathomechanisms of viral infections of the cardiovascular system, our knowledge regarding their treatment and management is still incomplete. Figure 6 displays some key features of cardiotropic viral infections, including the critical molecular structures by which viruses may possibly gain access to their cellular targets. Unless a cell is expressing ‘suitable’ cell surface receptors, cellular entry of the virus will not occur. Importantly, different viruses engage distinct receptors and this determines which specific cells within a tissue they are able to reach, i.e., their tropism within the host. Consecutively, tropism determines the development and progression of the disease. Accordingly, much effort is devoted to the elucidation of tropism. Thus, SARS-CoV-2 cellular entry involves binding to ACE2 receptor and cleavage by host cell surface protease TMPRSS2. Once entry occurs through the endo-lysosomal pathway, TLR- and RLR-dependent innate immune signaling is initiated, which subsequently triggers infiltration by diverse inflammatory cell populations. For further details, we refer the reader to excellent recent reviews summarizing current knowledge on viral infections of the heart, focused on pathophysiology, diagnostics, clinical relevance, and cardiovascular consequences, as well as current and emerging treatment strategies [163].

*Prophylaxis and vaccination:* Whereas conventional antiviral vaccine development methods have proven efficient against SARS-CoV-2, the most recent virus inducing the COVID 19 pandemic, novel RNA-based vaccines have yielded exceptionally good results against this pathogen. The revolutionary method successfully used to develop the BioNTecVC and ModernaVC vaccine was never before employed at scale, and indeed, the RNA modification/stabilization/purification methods, as well as the associated nanoparticle delivery tools, are of recent origin. Importantly, as emphasized by the authors of the landmark paper reporting the results of the BioNTecVC vaccine trial, development of the vaccination RNA sequence started immediately after publication of the novel virus genome sequence, which was derived soon after the recognition of COVID-19 as a new disease entity. Speed of development and adaptability to entirely new or variant viruses, which unfortunately are most likely to emerge in the future, bring significant advantages lasting beyond the current pandemic.

*Need for highly versatile antiviral tools*: The current pandemic, originating from transmission of a mutated animal virus to men, has heightened concerns and awareness that amongst the vast number of animal viruses, others may cross the species barrier to humans. Therefore, foresighted expansion of our antiviral arsenal appears warranted. Fortunately, novel therapeutic approaches as reviewed in Section 2.3 offer high versatility, enabling rapid adaption to essentially any coding or noncoding, viral or host cell, molecular target [176,177,178,179,180,181,182,183,184]. Further, their large-scale production will follow similar (i.e., RNA, DNA, and XNA) synthetic pathways, enabling massive up-scaling of therapeutics production if required.

*Advanced molecular virological tools:* While several human-pathogenic cardiotropic viruses are identified, there is good reason to believe that traditional molecular virological tools (e.g., PCR, RT-PCR) will fail to recognize novel viruses with any of those currently in focus. Thus, the etiology of giant cell myocarditis (GCM), most fulminant and life-threatening of the inflammatory cardiomyopathies, is unknown. GCM presents with extensive myocardial inflammation that only responds to high-dose immunosuppression. GCM has been associated with other autoimmune diseases, suggesting a relevant autoimmune component in its pathogenesis. However, the phenomenon of giant cells has been observed during viral infections such as herpes, suggesting a contribution of viral pathogens. In a recent paper, a study of plasma, peripheral blood mononuclear cells, endomyocardial biopsies (EMBs), and cardiac tissue samples from explanted hearts of patients with GCM and other subtypes of myocarditis [185], *Virome Capture Sequencing for Vertebrate Viruses (VirCapSeq-VERT)* was employed, a novel method that simultaneously screens for all known vertebrate viruses with sensitivity similar to real-time PCR [185,186]. The entire field of basic and clinical virology took great advantage from broad application of novel technologies and large—sometimes global—research consortia [187,188]. Within the cardiovascular field, extended use of these novel, more comprehensive virological tools may well lead to important insights into the pathogenesis of long-known but still etiologically enigmatic human diseases (cardiac sarcoidosis, eosinophilic cardiomyopathy, GCM, and others.)

### 3.2. The Human Genetic Architecture of SARS-CoV-2

The current COVID-19 pandemic, caused by infection with SARS-CoV-2, resulted in enormous health and economic burden worldwide [189,190,191,192]. One of the most remarkable features of SARS-CoV-2 infections is the extremely high variability of clinical sequelae, ranging from asymptomatic patients to life-threatening pneumonia and acute respiratory distress syndrome [163,170,171,172,173,193,194,195,196,197,198,199,200]. Since the rise of the COVID-19 pandemic, there has been an urgent need to identify pathophysiological characteristics leading to a severe clinical course in patients infected with SARS-CoV-2 [163]. 

Although established host factors correlate with disease severity (e.g., increasing age, male sex, higher body-mass index), these risk factors alone do not explain all of the variability in disease severity observed across individuals. Genetic factors contributing to COVID-19 susceptibility and severity may provide new biological insights into disease pathogenesis and identify mechanistic targets for therapeutic development or drug repurposing, as treating the disease remains a highly important goal despite the recent development of vaccines. A large number of genome-wide association studies (GWAS) addressing the contribution of common genetic variation to COVID-19 in different populations worldwide have provided support for the involvement of several genomic loci associated with COVID-19 severity and susceptibility [134,184,193,194,195,196,201,202,203,204,205,206,207]. The global COVID-19 Host Genetics Initiative (COVID-19 HGI) (on https://www.covid19hg.org/) (accessed on 31 October 2022) [208] recently reported the results of meta-analyses of 46 studies from 19 countries for host genetic effects [196]. Smaller studies analyzed, e.g., the association between COVID-19 severity and HLAs in 435 individuals from Germany (*n* = 135), Spain (*n* = 133), Switzerland (*n* = 20), and the United States (*n* = 147). This study described a biologically plausible potential association of HLA-C*04:01 with severe clinical course of COVID-19, as HLA-C*04:01 has fewer predicted binding sites for relevant SARS-CoV-2 peptides compared to other HLA alleles. 

For an excellent overview and discussion of invoked loci and their clinical implications, we refer the reader to van der Made et al. [201], who provide comprehensive tables with significant large-scale genome-wide associations in patients with severe or critical COVID-19. Importantly, they also review and discuss the reported outcomes of SARS-CoV-2 infection in patients with known inborn errors of immunity (IEI). One recent study [202] important for the assessment of GWAS emphasizes the impact of COVID-19 phenotype definitions, and revealed distinct patterns of genetic association and protective effects upon their replication analysis of 12 previously reported COVID-19 genetic associations. From a clinical practice perspective, GWAS have not identified a single gene locus with overwhelming impact upon disease course suggesting population-wide screening for high-risk individuals. On the other hand, for severely affected patients, genetic screening for IEI as suggested by van der Made et al. [201] may reveal individual clinical insights with possible therapeutic use.

Another type of contribution of large-scale molecular genetic analyses to better understand the variable clinical expression of SARS-CoV-2 infections, and the global dynamics of virus evolution, has been published by Chan et al. [134]. They have described the contrasting epidemiology and population genetics of COVID-19 infections defined by multilocus genotyping of the SARS-CoV-2 genomes. Their analysis of 22,164 SARS-CoV-2 genomes sampled worldwide suggests that the majority of SARS-CoV-2 infections in 2020 and 2021 were caused by genetically distinct variants that likely adapted to local populations. 

### 3.3. Cardiovascular Immunobiology of COVID-19 and Long COVID Syndromes

With regard to cardiovascular medicine, it is desirable to know specific genetic risk factors for the development of myocarditis during COVID-19 or upon vaccination [118,170,171,172,173,174,175,209,210,211,212,213,214], or for development of long COVID syndromes [195,215,216,217,218]. 

Fortunately, SARS-CoV-2 is rather rarely directly causing severe myocarditis or cardiomyopathies [170,171,172,173,174,175]. While SARS-CoV-2 may infect human engineered heart tissues and models of COVID-19 myocarditis [174], the observed effects in COVID-19 patients are rather induced by secondary immune phenomena than by the virus itself [171]. Since there is continuous molecular evolution of the SARS-CoV-2 virus, however, it is important to note that human endothelial cells have shown increased susceptibility to infections by SARS-CoV-2 variants [198]. In COVID-19 patients developing any myocardial involvement, the molecular virological and immunological tools outlined in Section 3.1 should be employed to characterize the myocardial disease, since this may enable individualized measures beyond standard heart failure and antiarrhythmic therapy. 

## 4. Novel Immune Pathomechanisms at Organ and Systemic Level

The individual response of the innate immune system to environmental (e.g., viral or other microbial infections) as well as to endogenous stimuli (e.g., tissue injury of any kind) is partially determined by genetic factors, but subject to modulation by non-genetic factors (e.g., stress of various types). While this has been well known for decades [219,220,221], recent research employing novel research tools [222,223,224,225,226,227,228,229,230,231,232] has uncovered interactions between brain and immune system at unprecedented resolution. Similarly, advanced molecular genetic methods [187,233,234,235,236,237,238] contributed to elucidate mechanistic pathways linking the GIT microbiome to the systemic innate immune response with its impact upon cardiovascular [239,240] and neurological diseases [241,242]. Below, we try to assess the clinical translational status of these research fields, focusing on therapeutic modulation of stress-induced disturbed brain–immune system interactions (Section 4.1), and of the GIT microbiome and its products (Section 4.2).

### 4.1. Brain–Immune System Interactions

The adverse effect of psychological stress upon various human diseases has been well known since decades [219,220,221] and several stress-induced brain–immune system interactions have been elucidated at cellular and molecular levels [220,243]. It is also obvious that stress reduction is highly desirable with regard to cardiovascular diseases [244,245,246,247], although often difficult to achieve in everyday life or the clinical setting. Any new avenue arising from recent neuroimmune research would be most welcome, of course. Clinically, neuromodulation strategies have been evaluated to reduce inflammation and lung complications of COVID-19 patients [248], and cardiovascular sequelae in posttraumatic stress disorders [249]. In experimental animal models, other approaches have been addressed: brain control of humoral immune responses by behavioural modulation [250]; modulation of the gut microbiome regulating psychological stress-induced inflammation [251,252]; IL-17A blockade or depletion of Th17 cell-inducing gut microbiota to reduce stress-induced vaso-occlusive episodes (VOEs) of sickle cell disease as a vascular disease model [251].

Folk wisdom has long suggested that emotional stress takes a toll on health. In the cardiovascular field, classical examples are acute myocardial infarction and Takotsubo syndrome [253]. The field of psychoneuroimmunology is now providing novel mechanistic insight into the pathways through which psychological stress and negative emotions are translated into physiological changes [221]. Neuroimmunology in general is one of the fastest-growing fields in the life sciences aiming to stepwise elucidate the highly complex interactions between nervous system and immune system at the molecular and cellular level [232,253,254,255,256]. It has been long known that acute and short-term stress induce rapid and significant redistributions of immune cells among different body regions. The underlying mechanisms are under close scrutiny, as stress-induced leukocyte redistribution appears to be of fundamental importance for survival. It appears critical to direct suitable immune cells to defined target organs in response to diverse external or internal challenges, thus significantly enhancing speed and efficacy of the immune response [232,254,255,256].

Despite this secured general knowledge, the details of the mechanistic pathways linking stress networks in the brain to peripheral leukocytes remain poorly understood. A recent experimental study has, for the first time, demonstrated that distinct brain regions shape leukocyte distribution and function throughout the body upon different types of acute stress in mice. Employing optogenetics and chemogenetics, this work revealed that motor circuits induce rapid neutrophil mobilization from the bone marrow to peripheral tissues via skeletal-muscle-derived neutrophil-attracting chemokines. Conversely, the paraventricular hypothalamus controlled monocyte and lymphocyte egress from secondary lymphoid organs and blood to the bone marrow through direct, cell-intrinsic glucocorticoid signaling. These stress-induced, counter-directional, population-wide leukocyte shifts were associated with altered disease susceptibility. On the one hand, acute stress changed innate immunity by reprogramming neutrophils and directing their recruitment to sites of injury. On the other hand, corticotropin-releasing hormone neuron-mediated immune cell shifts protected against the acquisition of autoimmunity, but impaired immunity against SARS-CoV-2 and influenza infection. These data identified distinct brain regions differentially and rapidly tailoring the immune cell landscape during psychological stress, thereby calibrating the ability of the immune system to respond to physical threats [232].

A recent study in mice and humans [256] revealed that sleep exerts lasting effects on hematopoietic stem cell function and diversity. In mice, sleep fragmentation altered the hematopoietic stem and progenitor cells (HSPCs) epigenome, priming cells for exaggerated inflammatory bursts. In humans, sleep restriction altered the HSPC epigenome and activated hematopoiesis. This work provides, for the first time, mechanistic insight into the prolonged effects of sleep disruption, another well-known stress factor.

A third neuro-immune study investigated whether brain activities may directly control adaptive immune responses in lymphoid organs [250]. It was found that splenic denervation in mice specifically compromised plasma cell formation during a T cell-dependent, but not T cell-independent, immune response. Neurons in the central nucleus of the amygdala (CeA) and the paraventricular nucleus (PVN) expressing corticotropin-releasing hormone (CRH) connected to the splenic nerve. Ablation or pharmacogenetic inhibition of these neurons reduced plasma cell formation, while pharmacogenetic activation of these neurons increased plasma cell abundance. A behaviour regimen, with mice made to stand on an elevated platform, led to activation of CeA and PVN CRH neurons and increased plasma cell formation. In immunized mice, the elevated platform regimen induces an increase in antigen-specific IgG antibodies. By identifying a specific brain–spleen neural connection autonomically enhancing humoral responses, and by demonstrating immune stimulation by behaviour modification, this experimental study revealed brain control of adaptive immunity, suggesting possible enhancement of immunocompetency by behavioural intervention.

Another study, employing retrograde tracing, and chemical as well as surgical and chemogenetic manipulations, identified a sympathetic aorticorenal circuit that modulates ILC2s in gonadal fat and connects to higher-order brain areas, including the paraventricular nucleus of the hypothalamus [257]. Similar to the other work, these results identify a neuro-mesenchymal unit translating signals from long-range neuronal circuitry into adipose-resident ILC2 function, thereby modulating host metabolism and obesity.

### 4.2. Immune Impact and Therapeutic Perspectives of the Intestinal Microbiome

The gastrointestinal tract (GIT) hosts a pool of immune cells representing 70% of the entire immune system, and the largest population of macrophages in the human body [250]. Through its local immune system, the GIT detects and responds to the local microbiome [242], but also impacts upon remote immune processes [258,259,260,261,262,263,264,265,266,267,268,269,270,271,272,273]. During the past years, multiple experimental studies have revealed that the microbiome and local immune system of the GIT may modulate distant inflammation within the cardiovascular system and brain [221,223,241,242,251,274,275,276,277,278,279,280,281,282]. However, whereas experimental models incriminate disturbed gut microbiota in a number of diseases (CNS disorders, atherosclerosis), data from human studies are sparse [257,276,277,283]. A theoretical basis for the use of microbiota-directed therapies in these disorders has been developed, but support from stringent clinical trials is missing and clinical confirmation is not yet received.

Regarding the cardiovascular system, a recent combined experimental and clinical study identified a novel regulatory circuit that links the gut microbiota metabolite propionic acid (PA), a short-chain fatty acid, with the gut immune system to control intestinal cholesterol homeostasis. The mechanism involves PA-mediated increase in regulatory T cell numbers and IL-10 levels in the intestinal microenvironment, subsequently suppressing the expression of NPC1L1, a major intestinal cholesterol transporter. In a proof-of-concept clinical study, it was demonstrated that oral supplementation of PA over the course of 8 weeks significantly reduced LDL and non-HDL cholesterol levels in hypercholesterolaemic subjects. The data suggest PA supplementation may improve cholesterol homeostasis and contribute to cardiovascular health. Another translational perspective is modulation of the gut microbiome by dietary approaches, or by prebiotics to sustainably increase the intestinal abundance of PA-producing species as an “intrinsic“ concept of atheroprotection [239,240]. An experimental study in mice found that a subset of integrin β7+ gut intraepithelial T lymphocytes within the small intestine enterocyte layer modulates systemic metabolism in a manner advantageous when food is scarce, but detrimental upon consumption of high fat and sugar diets [284]. This metabolic checkpoint might be therapeutically addressed by modulation of the GIT microbiome, e.g., through dietary approaches or prebiotics.

Critical impact of GIT microbiota upon the host immune system had previously been discovered in mice. Murine studies are paramount for the elucidation of basic biological phenomena, but have several limitations. These include conflicting results caused by divergent microbiota, and limited translational research value. Rosshart et al. [285] transferred C57BL/6 embryos into wild-type mice, thus creating "wildlings”. These had natural microbiota and pathogens at all body sites while retaining the well-defined and tractable genetics of the parent inbred strain. The bacterial microbiome, mycobiome, and virome [187,233,234,235,236,237,238] of “wildlings“ affected their immune landscape in multiple organs. Their gut microbiota outcompeted lab strain microbiota and proved resilient to environmental challenge. “Wildlings“, but not the lab mice, phenocopied human immune responses in two preclinical investigations. This landmark study demonstrated that a combined natural microbiota- and pathogen-based model holds promise to enhance the reproducibility of experimental biomedical research and to improve translational success of immunological studies.

Regarding the brain, a series of important experimental studies uncovered hitherto unsuspected bidirectional relationships between gut and brain, challenging the classical view of the central nervous system as an immune-privileged site. These studies identified new specialized immune cell subtypes located to distinct anatomical sites: skull and vertebral bone marrow are myeloid cell reservoirs for meninges and CNS parenchyma [286]; the meninges contain diverse immune cells populations: macrophages, T cells, B cells [287], plasma cells producing IgA essential for defense of the central nervous system [288], and gut-licensed NK cells driving anti-inflammatory astrocytes [289]. Furthermore, host GIT microbiota constantly control maturation and function of microglia in the CNS [290,291,292,293,294]. They are also capable of reversibly modulating behavioural and physiological anomalies associated with neuroinflammation [285,295,296,297]. 

Figure 7 provides a synopsis of the novel players in CVD innate immunity discussed so far (noncoding genome/epigenome, cardiotropic viruses, brain–immune system and gut–brain–immunity axis).

## 5. Clinical Perspectives of Recent Studies into Novel Immune Mechanisms

### 5.1. Recognition of High Genetic Risk for Severe COVID-19 or Cardiovascular Involvement

One of the most vexing features of SARS-CoV-2 infections [298] is the extraordinary variability of clinical consequences, ranging from asymptomatic to pneumonia and acute respiratory distress syndrome. Of course, variable clinical course is common in diverse viral diseases, but rarely to such an extent as observed with SARS-CoV-2. Established host factors such as high age or body-mass index or concomitant diseases correlate with disease severity, but do not explain all of the variability observed across individuals. These observations have driven most extensive research into genetic factors contributing to COVID-19 susceptibility and severity [163,170,171,172,173,193,194,195,196,197,198,199,200], and the wish to identify otherwise healthy individuals at high genetic risk for severe COVID-19 will persist, whatever progress regarding prophylaxis or treatment options is made. 

From the perspective of individualized medicine, recognition of particularly high individual risk may enable tailor-made approaches towards these individuals, even when the disease is already ongoing. In that regard, van der Made et al. [201] have emphasized the often critical outcomes of SARS-CoV-2 infection in patients with defined inborn errors of immunity which confer specifically addressable weaknesses of the immune defense.

### 5.2. Inflammation as Therapeutic Target in Life-Threatening Arrhythmias and Heart Failure

Anomalous immune activation and macrophages are involved in multiple types of cardiomyopathies. From a clinical perspective, there are multiple obstacles to unequivocal detection of myocardial inflammation as a precipitating factor for life-threatening arrhythmias or cardiac remodeling and failure. Even in narrowly focused patient cohorts for whom genetic predisposition is suspected—such as by detection of pathogenic variants— this requires significant additional diagnostic work-up (cardiac MRI, EMB, PET-CT), which may not be feasible in all cases. It is highly likely that only a small fraction of all inflammation-triggered arrhythmic events or progressive heart failures will be clinically detected, generating a blind spot regarding the potential of anti-inflammatory treatments in this context. This deficiency strongly suggests consistent interdisciplinary translational work taking advantage of state-of-the-art genetics, immunology, and clinical cardiology, in particular in younger patients presenting with unexplained symptoms and signs. These are most likely to benefit from thorough examination to enable delay of disease progression or even cure.

Notably, anti-arrhythmic potential of immunosuppression does not rely on genetic SCN5A variants because inflammation per se may cause dysfunction of normal SCN5A channels, generating a broader clinical incentive to follow this line of research. In view of possible therapeutic relevance, it appears recommendable to further evaluate the hypothesis of independent immunologic risk in major SCN5A variant-carrier cohorts [25]. For further in-depth discussion of recent approaches towards the potential of immunomodulating therapies in human cardiomyopathies, we may refer the reader to a recent review covering the field from classical inflammatory cardiomyopathies to immune-checkpoint inhibitor-associated and SARS-CoV-2-associated myocardial inflammation [29].

### 5.3. Remaining Clinical Challenges in Gastrointestinal Microbiome and Neuroimmune Stress Research

The novel immune pathomechanisms at organ and systemic level discussed in Section 4 have certainly opened new avenues for clinical research. Experimental studies have documented significant impact of the gut microbiome and gut–brain–axis upon remote inflammation in cardiovascular system and brain. The rationale for the use of microbiota-directed therapies in these disorders is obvious; a clinical pilot trial was recently published [240] but final support from stringent clinical trials is not yet available. 

Within the field of psychoneuroimmunology, recent landmark experimental studies employing advanced genetic and neurosurgical methods have provided novel mechanistic insights into adverse immune impact resulting from different types of stress. Remarkable is their elucidation of differential impact arising from defined brain areas and neuroimmune pathways. These recent methodological developments are likely to enable significantly deeper insight into long-known classical clinical phenomena, such as stress-induced acute myocardial infarction or Takotsubo syndrome [253]. Interdisciplinary work involving neurology, immunology, and cardiology will increasingly exploit the new analytical tools. Translational evaluation of their possible relevance in clinical cardiovascular medicine is eagerly awaited.

Figure 8 critically summarizes the clinical translational status of insights from the novel research fields, classified from experimental animal studies, through clinical research and pilot trials, to large-scale clinical trials. Among the research fields, molecular genetics has made considerable contributions to improve clinical diagnostics and therapy within and beyond cardiovascular medicine.

## 6. Future Directions for Interdisciplinary Translational and Clinical Research

While multiple excellent recent reviews have addressed diverse important aspects of immunological research into cardiovascular disorders, we have deliberately focused on selected studies characterized by their use of particularly advanced methodology in preclinical or clinical settings. Due to the immense worldwide efforts in the field during the past years, the present review cannot be comprehensive, but instead tries to convey an up-to-date perspective on promising developments which may shape research at the crossroads of cardiology–immunology–neurology. 

During the next few years, extensive use of the new research tools should lead to a deeper understanding of the processes at these crossroads, long recognized, but still far from being fully exploited in clinical medicine. Although several challenges need to be overcome before the full impact of these far-reaching new findings will hit the clinical arena, the above-reviewed studies already exemplify an overarching aspect, i.e., the interdisciplinary character of work to come. 


**Key Messages**


Practice-ready affordable advanced genetic diagnostics has entered clinical practice and is continuously providing important insights, not only into an individual’s genetic risk, but perhaps also in uncovering new pathomechanisms suitable for individualizing therapy.After decades of experimental and translational work, progress from traditional pharmacological towards nucleic acid-based therapies for cardiovascular diseases has been achieved. Insights from basic genetic research (RNA interference, antisense drugs, CRISPR-Cas) are emerging as fruitful for clinical medicine.Fundamentally improved understanding of the intestinal immune system and microbiome with experimental evidence for far-reaching immune impact upon cardiovascular system and brain has triggered clinical trials evaluating the potential of microbiome modulation.The rapidly evolving field of neuroimmunology has identified novel brain–immune system interaction networks revealed at unprecedented resolution, and documented grave adverse impact of stress upon cardiovascular and virological diseases.

## Figures and Tables

**Figure 1 jcm-12-00335-f001:**
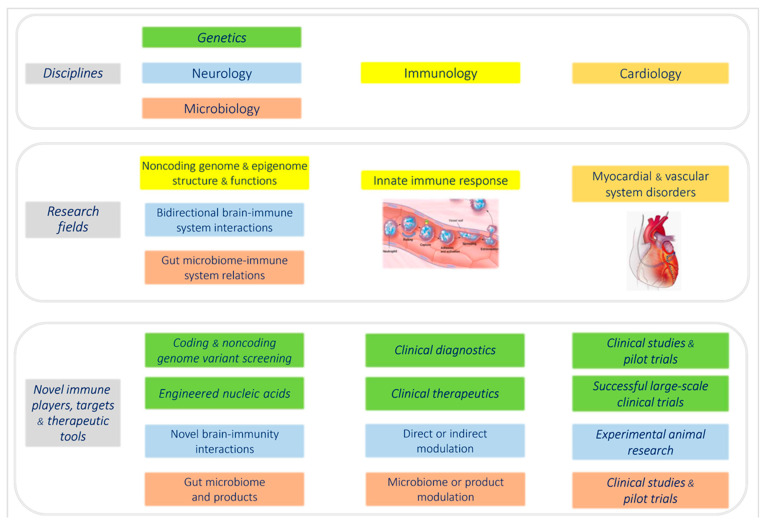
New research areas for cardiovascular disease immunity. Several rapidly expanding research fields investigating cardiovascular innate immunity require logistic and advanced technological cooperation between the disciplines of genetics, neurology, microbiology, and immunology. Interdisciplinary use of powerful new research tools should lead to deeper understanding of the processes governing cardiovascular immunopathogenesis. We try to convey a perspective on promising developments at the crossroads of cardiology–immunology–neurology. Whatever the ultimate clinical impact of research in these fields, an obvious overarching aspect is the interdisciplinary character of work to come.

**Figure 2 jcm-12-00335-f002:**
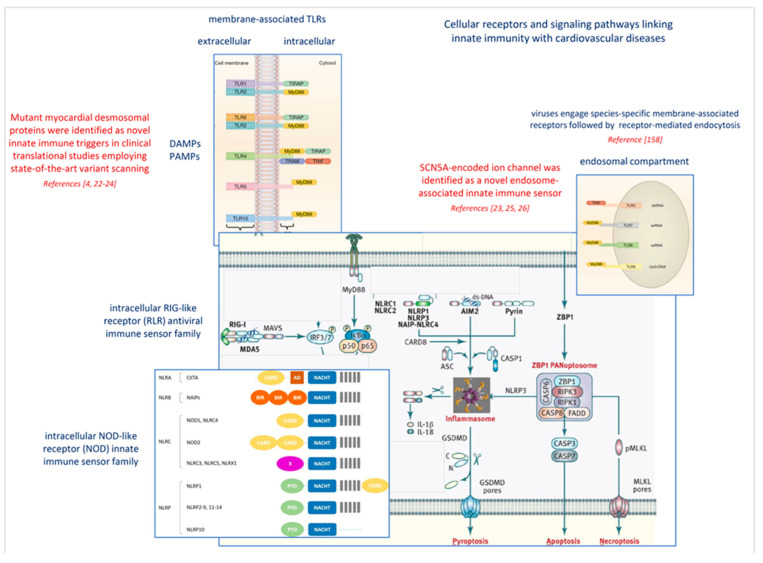
Cellular receptors and signaling pathways linking innate immunity with cardiovascular diseases. Pattern recognition receptors (PRRs) play a critical role in the activation of an innate immune response to invading pathogens, e.g., viruses (see chapter 3 below) or mutant or otherwise modified endogenous structures. PRRs enable recognition of pathogen-associated molecular patterns (PAMPs), or of endogenous molecules released from injured cells and tissue, designated damage-associated molecular patterns (DAMPs), acting as PRR ligands. Ligand–PRR interaction then triggers intracellular signaling pathways, ultimately inducing expression and release of a multitude of pro-inflammatory or antiviral cytokines. PRRs may be associated with the cell membrane (e.g., 10 Toll-like receptors—TLRs; C-type lectin receptors—CLRs), located within the cytosol (23 NOD-like receptors—NLRs; 3 three RIG-like receptors—RLRs), or in specialized intracellular compartments such as endosomes particularly relevant in viral infections (TLRs 3, 7, 8, 9). These PPRs are characterized by a variable number of ligand-sensing receptors (LRR) at their N-terminal (TLRs) or C-terminal (NLRs) ends, and one or more protein–protein interaction (TIR, CARD, PYR) or oligomerization (NACHT) domains. NLRs are also involved in the formation of inflammasomes, a molecular machine activating inflammatory processes including programmed cell death. While our understanding of the highly complex multiscale signaling network of innate immunity [22] is incomplete, a number of components have been elucidated in considerable detail. Here we may refer to excellent reviews addressing PRRs of particular interest for cardiovascular medicine [20,23] (TLRs, NLRP3–inflammasome pathway, IL-1 to IL-6 pathway).

**Figure 4 jcm-12-00335-f004:**
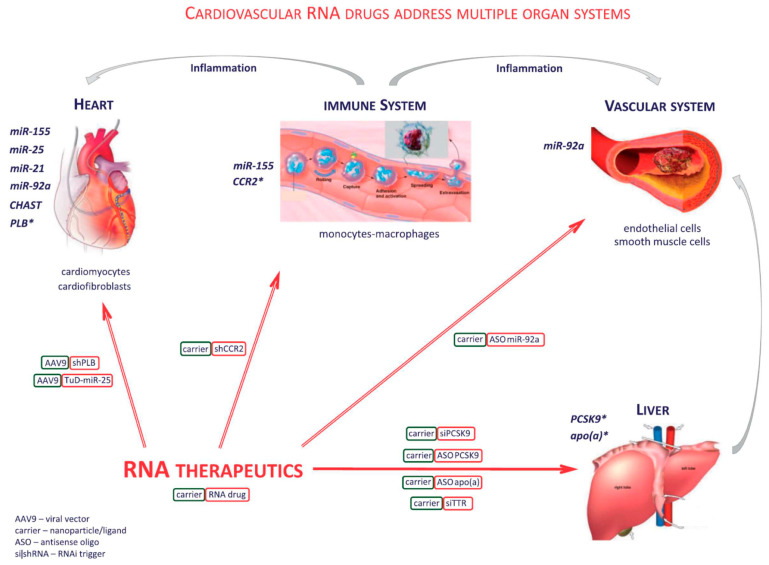
Novel nucleic acid therapeutics targeting conventional protein-coding genes or noncoding RNA targets (reprinted from Poller et al. Eur. Heart. J. 2018 [1] by permission). Cardiovascular RNA drugs address multiple organ systems. RNA drugs do not need to address heart (A) or vasculature (B) directly. Instead, primarily liver-targeted RNA drugs (C) are the currently most successful development in the cardiovascular field. Another group of strategies addresses the immune system (D), in particular monocytes-macrophages. ’Carrier’ denotes a synthetic nanoparticle and/or receptor ligand employed to deliver an RNA drug to its tissue target. ‘Carrier’ is bound to and serves to stabilize the RNA drug within the circulation, and to endow it with at least partial selectivity for the target cells, in order to minimize side effects. ‘AAV9’ denotes a cardiac-targeting recombinant adeno-associated viral vector containing a genome from which the therapeu-tic RNA sequence is continuously transcribed. Molecular targets: apo(a), apolipoprotein (a); CCR2, chemokine C-C motif receptor 2; CHAST, Cardiac hypertrophy associated transcript; PCSK9, proprotein convertase subtilisin/kexin type 9; PLB, phospholamban. (*modified from [1] by permission of Eur. Heart J*.).

**Figure 5 jcm-12-00335-f005:**
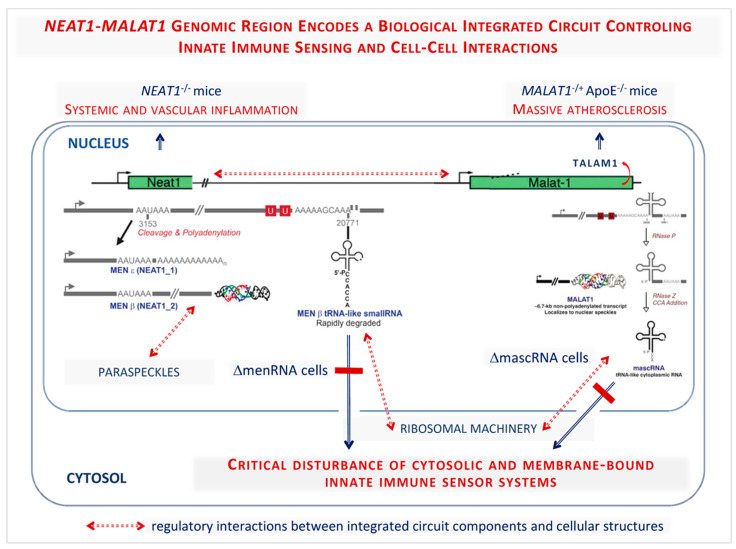
The *NEAT1-MALAT1* genomic region encodes a biological integrated circuit controlling innate immune sensing and cell–cell interactions (reprinted from Gast et al. Cells 2022 [131] by permission). From an evolutionary perspective, the *NEAT1-MALAT1* genomic region appears as a highly integrated RNA processing circuitry critically contributing to immune homeostasis. Its components MEN-β, MEN-ε, menRNA, *MALAT1*, *TALAM1*, and mascRNA are obviously set for well-balanced interactions with each other. Genetic ablation of any element therefore leads to major dysfunction. Beyond prior work in *NEAT1* and *MALAT1* knockout mice, a recent cell biological study identified menRNA and mascRNA as novel components of innate immunity with deep impact upon cytokine regulation, immune cell–endothelium interactions, angiogenesis, and macrophage formation and functions. These tRNA-like transcripts appear to be prototypes of a class of ncRNAs distinct from other small transcripts (miRNAs, siRNAs) by biosynthetic pathway (enzymatic excision from lncRNAs) and intracellular kinetics, suggesting a novel link for the apparent relevance of the *NEAT1-MALAT1* cluster in cardiovascular and neoplastic diseases. (*Modified from [131] by permission from Cells*).

**Figure 6 jcm-12-00335-f006:**
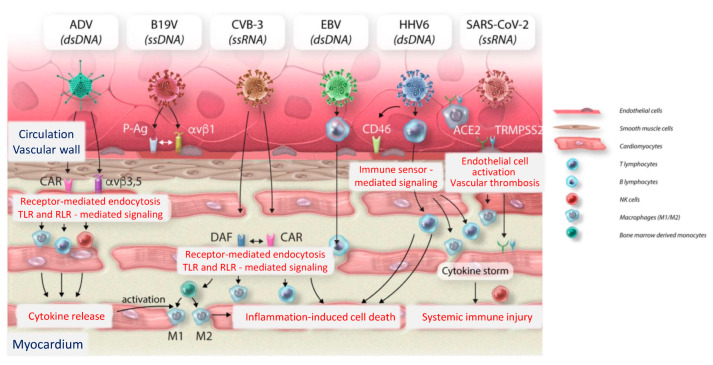
Cellular receptor-mediated entry and innate immune activation mechanisms of human-pathogenic viruses (modified from Cardiovasc. Res. 2022 [163] by permission). Depicted are the most abundant cardiotropic viruses, as well as SARS-CoV-2, and their molecular entry mechanism into cardiovascular cells in the heart. CVB-3 and ADV enter cardiomyocytes via binding the transmembrane CAR. In addition, decay-accelerating factor serves as CVB-3 receptor. Integrins (αvβ3 and αvβ5) promote ADV internalization. B19V targets endothelial cells by binding to erythrocyte P antigen and integrin αvβ1 as co-receptor. EBV efficiently infects resting human B lymphocytes, whereas HHV6 primarily targets CD4+ T lymphocytes. Using CD46 as cellular receptor, HHV6 can directly infect endothelial cells and subsequently enter adjacent tissues. SARS-CoV-2 cellular entry involves specific binding to the ACE2 receptor, as well as proteolytic cleavage by the host cell surface serine protease TMPRSS2. For SARS-CoV-2, several cardiac targets including vascular endothelial cells and cardiomyocytes are proposed. Moreover, pulmonary-derived macrophages are suggested, carrying the virus into the myocardium. As a consequence of receptor-dependent cellular entry through the endo-lysosomal pathway, a TLR innate immune signaling cascade is initiated, followed by infiltration by inflammatory cells (T and B lymphocytes, natural killer cells, bone-marrow derived monocytes) which differentiate into M1 and M2 macrophages. ds, double stranded; ss, single-stranded. *(Modified from [163] by permission of Cardiovasc. Res.).*

**Figure 7 jcm-12-00335-f007:**
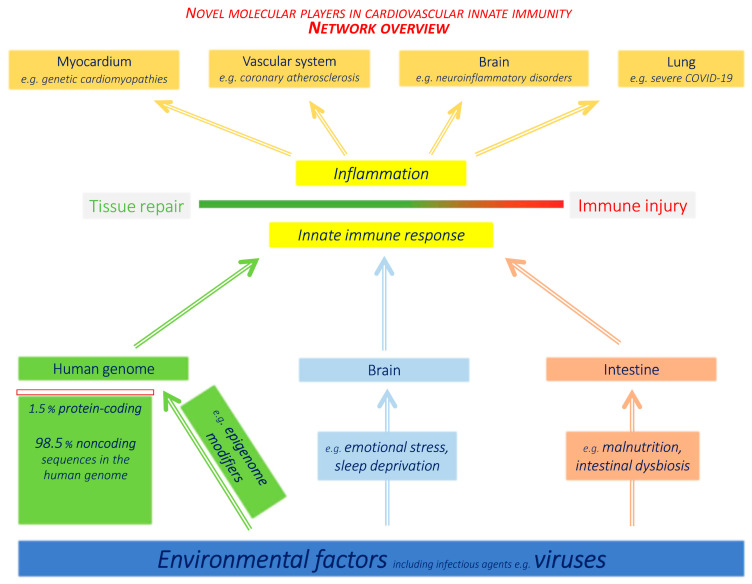
Novel molecular players in cardiovascular innate immunity—an overview of systems involved. Several systems acting through innate immunity upon cardiovascular diseases have come under intense scrutiny, taking advantage of advanced research tools. (1) Novel molecular components of the human noncoding genome and epigenome emerged as modulators of immunity acting through non-canonical pathways. (2) The intestinal immune system and microbiome exert a far-reaching impact upon distant tissues (cardiovascular system, brain). (3) Brain–immune system interaction networks are becoming revealed at unprecedented resolution. (4) Classical environmental risk factors, e.g., psychosocial stress or viral infections, are under intense investigation, again triggered by the SARS-CoV-2 pandemic with major direct and indirect impact upon cardiovascular health. In all of these fields, recent insights into human molecular genetics, as well as advanced genetic research tools, made major contributions.

**Figure 8 jcm-12-00335-f008:**
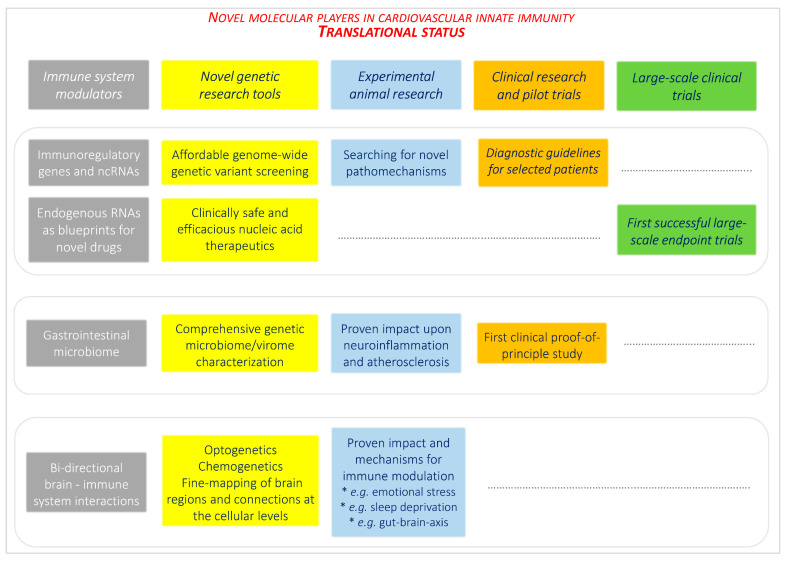
Clinical translational status of the research fields.

## Data Availability

Not applicable.

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
