# Peer review of "Innate Immunity in Cardiovascular Diseases—Identification of Novel Molecular Players and Targets"

_jcm, 2023, doi:10.3390/jcm12010335_

Round 1

Reviewer 1 Report

The review from Poller et al focuses on the role of inflammation in human cardiomyopathies with particular attention to multi organ interconnections.

While the review is well written and the reader is well guided into the different paragraphs, the contents are highly generals with many references to other reviews. The review would benefit from the addition of some extra details.

In paragraph 2.2, the authors describe the importance of Foxo3 SNP in regulating immune response. Would it be possible to integrate this information with the role of Foxo3 in cardiovascular diseases (CVDs) and to provide more details on the correlation between Foxo3 SNP in CVDs and immune regulation?

The figures are well done, clear and filled of relevant information. It would be nice to insert in the main text some of the information provided only in the figures. In particular, figures 2-3-4-5 contain important information that if present in the main text would provide a more comprehensive view to the reader.

In paragraph 2.3, the authors write “each lncRNA needs to be analyzed per se because of their diversity, and therefore suggest to focus research upon species fulfilling diverse criteria of biological relevance in humans”. Would it be possible to expand this concept?

The authors introduce the really recent described and interesting role of mascRNA and menRNA in immune functions. Due to their function on foam cell formation, would it be possible to speculate more on their role in the onset or progression of CDVs such as atherosclerosis?

Paragraph 4 gives an overview on the brain – immune interaction and their effects on the cardiovascular compartment, with particular focus on how the brain regulates the immune compartment. The Takotsubo syndrome is a clear example of the brain - immune compartment – heart interconnection. A brief but comprehensive description of this pathology would complete this paragraph.

Minor points:

In paragraph 2.1, the abbreviations ICDs and EMB are used. Please state the non-abbreviated form.

Remove “NEW” in front of Figures 2,3,4,5.

Author Response

We submit our reply to Reviewer 1 with the attached file

Reviewer 2 Report

Title: Innate immunity in cardiovascular diseases – Identification of novel molecular players and targets.

In this manuscript, the authors focused on diverse aspects of immunological research in cardiovascular diseases to identify novel molecular players.

 The manuscript is well written.  

Author Response

We submit our reply to Reviewer 2 with the attached file

Reviewer 3 Report

Poller et al. describe the molecular players and innate immune targets in cardiovascular disorders in their review. Overall, the review is well-organized and emphasizes the potential for novel molecular players and targets in cardiovascular innate immunity. The review requires revision to meet the high quality criteria of the journal, despite the reality that it is an interesting review on a highly intriguing topic.

1. There are many grammatical mistakes throughout the review. The text should be thoroughly proofread for grammar errors.

2. There are many errors and punctuation marks, for example: Many of the (-) symbols in the abstract are nonfunctional, also as is Line 111 (it is and is not it is), Line 527 "wildlings." 

3.  Fig. 2 and Fig. 3's resolutions are not enough, and many details are not clear.

4.  Permissions from the journal where the illustration was published should be shown to the Journal of Clinical Medicine.

5. The title and subtitles "Clinical perspectives of recent studies into new immune mechanisms" should be talked about in depth. 

6. New title "7. Future Directions" should be addressed.

Author Response

We submit our reply to Reviewer 3 with the attached file
